# Fatigue Life Prediction of Reinforced Concrete Beams Strengthened with CFRP: Study Based on an Accumulative Damage Model

**DOI:** 10.3390/polym11010130

**Published:** 2019-01-13

**Authors:** Xin-Yan Guo, Yi-Lin Wang, Pei-Yan Huang, Xiao-Hong Zheng, Yi Yang

**Affiliations:** 1School of Civil Engineering and Transportation, South China University of Technology, Guangzhou 510640, China; xyguo@scut.edu.cn (X.-Y.G.); 15086715590@163.com (Y.-L.W.); xhzheng@scut.edu.cn (X.-H.Z.); yiyang@scut.edu.cn (Y.Y.); 2State Key Laboratory of Subtropical Building Science, South China University of Technology, Guangzhou 510640, China

**Keywords:** fatigue life prediction, prestress, carbon fiber reinforced polymer (CFRP), accumulative damage model

## Abstract

With the prestressed carbon fiber reinforced polymer (CFRP) strengthening technique widely used in reinforced concrete (RC) structures, it is more and more important to study the fatigue performance of RC structures. Since the fracture of a tensile steel bar at the main cracked section is the leading reason for the failure of RC beams reinforced by prestressed CFRP, a fatigue life prediction model of RC beams reinforced by prestressed CFRP was developed based on an accumulative damage model. Moreover, gradual degradation of the performance of the concrete was considered in the fatigue life prediction model. An experimental study was also conducted to research the fatigue behavior of RC beams reinforced by prestressed or non-prestressed carbon fiber laminate (CFL). During the tests, fatigue crack patterns were captured using a digital image correlation (DIC) technique, and the fatigue lives of a total of 30 beams were recorded. The results showed that the predicted main crack propagation curves and the fatigue lives were close to the experimental data. This study also exhibited that the prestressed CFRP could reduce the stress of main steel bars in RC beams and effectively improve the fatigue performance of the RC beams.

## 1. Introduction

Bonding carbon fiber reinforced polymer (CFRP) onto the surface of reinforced concrete (RC) is a lightweight, efficient, and noncorrosive strengthening technique and has become prevalent in the past two decades [1,2]. However, the disadvantage of the non-prestressed CFRP reinforcement technology is that the reinforcement strength of CFRP is not fully utilized. To overcome this disadvantage, an active reinforcing technique using prestressed CFRP to strengthen RC beams has been developed by researchers.

Compared to the non-prestressed CFRP strengthening technique, the prestressed CFRP strengthening technique has some advantages [3,4,5]: improving the stress distribution of concrete, limiting the propagation of cracks, utilizing the high tensile strength of CFRP in a preferable way, closing existed cracks, and increasing the fatigue lives of the strengthened structures.

Currently, with the development of prestressed CFRP reinforcement technology, more and more researchers are paying attention to the fatigue performance of the RC beams strengthened with prestressed CFRP. Some researchers [6,7] carried out experiments to study the behavior of RC beams strengthened with prestressed CFRP under fatigue loads. EI-hacha [8] showed that the fatigue performance of the specimens could be greatly improved by prestressed CFRP plates and that the anchoring methods also had an important influence. As discussed by Xie [9], the stresses of the steel bars and the fatigue damage were further reduced and the fatigue lives of RC beams were increased by prestressed CFRP. Huang [10] conducted the fatigue experiments on RC beams strengthened with prestressed CFRP and observed that the main failure modes were fatigue fracture of the main steel bar and debonding failure of the CFRP-concrete interface.

On the basis of experimental study, some researchers found that the stress amplitude of tensile steel bars was the main factor affecting the fatigue lives of the beams and that prestressed CFRP could reduce the stresses of the tensile steel bars and ultimately increase the fatigue lives of RC beams. Huang et al. [11] proposed a semi-empirical formula for predicting the fatigue lives of RC beams reinforced by prestressed CFRP under cyclic loading. According to the fatigue stress amplitude of the main steel bars measured during tests, the fatigue lives were predicted by the formula. Fadi et al. [11] developed a fatigue life prediction model that considered the effect of prestressed CFRP reinforcement on the fatigue behavior. In the model, the compatibility equations of RC beams strengthened with prestressed CFRP rest on an assumption of a linear relationship. Xie et al. [9] established a fatigue accumulative damage model based on the degradation of flexural stiffness to predict the fatigue lives of the strengthened beams. H. Huang [12] conducted fatigue experiments on RC beams strengthened with prestressed CFRP. It was found that the fracture of the tensile steel bar at the main cracked section was the main failure mode of prestressed CFRP-strengthened RC beams under fatigue loads. Miner’s rule can be used to predict the fatigue lives of CFRP for strengthening RC beams until the fracture failure of the tensile steel bar occurs. X. Guo [13] proposed a modified version of Paris’ law for predicting fatigue life of RC beams strengthened with prestressed CFRP. In this model, *J*-integral was calculated by using the finite element method. H. B. Park [14] carried out fatigue test of RC beam strengthened with prestressed FRP tendon. From the test results, the steel and FRP strains were described according to the accumulation of fatigue loading. Some researchers found the fatigue behavior of composites follow the Weibull distribution. Moreover, a probabilistic model of fatigue failure was provided [15]. Based on numerical calculations by an open source program—ProFatigue^®^, Hanif [16] provided failure fatigue life assessment at various probabilities.

Although some aspects of the fatigue performance of prestressed CFRP for strengthening RC beams have been explored in previous models, the gradual degradation of the performance of the concrete has not been considered. Holmen’s experimental results [17] indicated that the compressive stress-strain relationship of concrete changes continuously, owing to the internal fatigue damage accumulation in the concrete. The time-dependent constitutive relationship of concrete has an impact on stresses in tensile steel bars. The fatigue behavior of RC beams can be accurately described by considering the degradation of the performance of concrete in the stress-strain relationship.

In view of the above considerations, the emphases of this paper are to establish a fatigue life prediction model based on the accumulative damage to tensile steel bars and to consider gradual degradation of the performance of concrete. To verify the model, the fatigue lives of a total of 30 RC beams strengthened with CFRP which underwent different levels of prestressing (0%, 10%, 15%, and 22%) were compared with theoretical values. The theoretical crack evolutions for 13 specimens in different cycles were also compared with testing results. The difference between the models with and without consideration of the gradual degradation of the performance of the concrete was analyzed.

## 2. Fatigue Life Prediction Method

As widely observed from the fatigue tests on RC beams strengthened without prestressed CFRP in earlier times, the fracture of tensile steel bars at the main cracked section is the controlling failure mode [18]. Heffernan and Erki [19] found that CFRP shared part of the stress with the main steel bars in RC beams and improved the fatigue performance of beams. This behavior has also been found in RC beams strengthened with prestressed CFRP [20]. Therefore, the fatigue lives of RC beams reinforced with prestressed CFRP or without prestressed CFRP can be obtained by the fatigue lives of the tensile steel bars. In the cracked section, a new method to analyze the fatigue lives of RC beams strengthened with prestressed CFRP was proposed based on Miner’s rule [21]; a fatigue life prediction method for non-prestressed CFRP-reinforced beams has been previously developed. In addition, the gradual degradation of the performance of the concrete was considered in the fatigue life prediction.

### 2.1. Fatigue Damage of Tensile Steel Bars

Miner’s rule has always been used to calculate the accumulated fatigue damage of the tensile steel bars:(1)$$D=\sum \frac{n_{i}}{N_{i}},$$
where, *D* is the accumulation of fatigue damage (*D* ≤ 1); *n_i_* is the cycle number for the specified stress amplitude $\sigma_{si}$ of the tensile steel bars; and $N_{i}$ is the final cycle number to failure for the stress amplitude $\sigma_{si}$ of the tensile steel bars.

Many fatigue life models describing the fatigue behavior of the steel bars have been published. Wang [22] described these equations and compared them with experimental results from the literature. He found that the existing models did not agree well with the experimental data over the whole course of fatigue, and he conducted a regression analysis based on the scarcity of published data on the fatigue lives of RC beams. Considering the models constructed by Oudah [23], Diab [24], and Moss [25], three regions were combined to describe the S-N relationship of the steel bars. The proposed relationship can be given by the following equation:(2)$$S_{si}=\;\{\begin{array}{c} 797.9-94.31\log N_{i}\;\mathrm{for}\;1\;<N_{i}\leq 10^{4}\\ 458.8-9.566\log N_{i}\;\mathrm{for}10^{4}<N_{i}\leq 10^{5}\\ 1208-159.4\;\log N_{i}\;\mathrm{for}10^{5}<N_{i}\leq 10^{6}\\ 601.3-58.45\log N_{i}\;\mathrm{for}10^{6}<N_{i}\leq 10^{7} \end{array},$$
where *N_i_* is the cycle number to fatigue failure and *S_si_* is the applied stress range in MPa.

Equation (2) can also be expressed as follows:(3)$$\log N_{i}=\;\{\begin{array}{c} 8.460-0.011S_{si}\;\mathrm{for}420.5<S_{si}\leq 787.7\\ 47.96-0.105S_{si}\;\mathrm{for}411.0<S_{si}\leq 420.5\\ 7.578-0.006S_{si}\;\mathrm{for}251.6<S_{si}\leq 411.0\\ 10.29-0.017S_{si}\;\mathrm{for}192.2<S_{si}\leq 251.6 \end{array},$$

For the fatigue failure mode of the steel bars, the fatigue lives of RC beams strengthened with prestressed CFRP or without prestressed CFRP can be predicted from accumulative damage until the tensile steel bar fractures (*D* = 1).

### 2.2. Elastic Modulus Degradation of Concrete

As the number of loading cycles increases, the stress amplitude of the tensile steel bar changes because of the degradation of the concrete performance and the propagation of the cracks.

In the fatigue testing, steel bars in the RC beams strengthened with CFRP maintained a linear elastic state before yielding. At the same time, because CFRP has excellent fatigue resistance, it maintained a linear elastic state over its whole fatigue life. Due to the stress range of tensile steel bars being the main factor affecting the fatigue lives of the beams, this study places an emphasis on changes in the stress range of the tensile steel bars caused by elastic modulus degradation of the concrete.

With repeated fatigue loading, the elastic modulus of concrete changes due to accumulative damages [26]. EI-Tawil [27] proposed a formula for the effective elastic modulus of concrete after *n_i_*—many loading cycles as follows:(4)$$E_{ci}=\left(1-0.33\frac{n_{i}}{N_{ci}}\right)E_{c},$$
where *E_ci_* is the effective elastic modulus of concrete at the *i*th cycle and *E_c_* is the initial elastic modulus of concrete. *N_ci_* is the number of loading cycles to failure for concrete, which can be calculated using the following equation [28]:(5)$$S_{ci}=0.9885-0.0618\mathrm{lg}N_{ci},$$
where, *S_ci_* is maximum stress level at *i*th cycle and *S_ci_* = σ*_ci_*/*f_c_*, where $\sigma_{ci}$ and *f_c_* are maximum compressive stress and uniaxial compressive strength of concrete, respectively.

### 2.3. Time-Dependent Constitutive Relationships of the Tensile Steel Bar

In this study, the following constitutive model of the tensile steel bars was applied [29]. Under monotonic loading, the yield, hardening and softening phenomena of the steel bars can be accurately described as follows:(6)$$\sigma_{si=}\{\begin{array}{c} E_{s}\varepsilon_{si}\;\varepsilon_{si}<\varepsilon_{y}\\ f_{y}\;\varepsilon_{y}<\varepsilon_{si}<k_{1}\varepsilon_{y}\\ k_{3}f_{y}+\frac{E_{s}\left(1-k_{3}\right)}{\varepsilon_{y}{\left(k_{2}k_{1}\right)}^{2}}{\left(\varepsilon_{si}-k_{2}\varepsilon_{y}\right)}^{2}\;\varepsilon_{si}>k_{1}\varepsilon_{y} \end{array},$$
where *E_s_* and *f_y_* represent the elastic modulus and yield strength of the tensile steel bar, respectively; $\varepsilon_{si}$ is the longitudinal strains of the tensile steel bar at the *i*th cycle; and $\varepsilon_{y}$ is the strain value corresponding to the yield strength of the tensile steel bar. As shown in Figure 1, the values of *k*_1_, *k*_2_, *k*_3_, and *k*_4_ are chosen according to the type of tensile steel bar.

### 2.4. Equilibrium and Compatibility Equations

For RC beams strengthened with prestressed CFRP or without prestressed CFRP, the stress amplitude of the tensile steel bars would change with increasing number of loading cycles according to the degradation of the concrete performance and the generation and propagation of cracks. With the sectional analysis method, the crack height *a_i_* and the location *h_ci_* of the neutral axis at the *i*th cycle could be determined with the assumption of a linear strain distribution, as shown in Figure 2. The compressive stress distribution of concrete is parabolic stress block [30]. The crack height *a_i_* and the location *h_ci_* of the neutral axis would change with the effective elastic modulus *E_ci_* of the concrete at the *i*th cycle. Based on the sectional equilibria of forces and moments, the following equations can be formed:
(7)$$\{\begin{array}{c} \frac{1}{2}bh_{ci}f_{t}+\sigma_{si}A_{s}+E_{f}A_{f}\left(\varepsilon_{fi}+\varepsilon_{pe}\right)=E_{s}^{\prime }A_{s}^{\prime }\varepsilon_{s}^{\prime }+\int_{0}^{y_{ci}}E_{ci}\varepsilon_{ci}bdy\\ M=\frac{1}{2}bh_{ci}f_{t}\cdot \frac{2}{3}h_{ci}+\sigma_{si}A_{s}\left(a_{i}-c+h_{ci}\right)+E_{f}A_{f}\left(\varepsilon_{fi}+\varepsilon_{pe}\right)\left(a_{i}+h_{ci}\right)+E_{s}^{\prime }A_{s}^{\prime }\varepsilon_{s}^{\prime }\left(h-a_{i}-h_{ci}-c^{\prime }\right)+\int_{0}^{y_{ci}}E_{ci}\varepsilon_{ci}bydy, \end{array}$$
where *M* is the bending moment at the main cracked section; *b* and *h* are the width and height of the RC beam; *y_ci_* and *h_ci_* are the heights of the compression and tension zones, respectively, for the concrete at the *i*th cycle at the main cracked section; *a_i_* is the height of main crack at the *i*th cycle; *c′* and *c* are the depths of the concrete cover; *E_f_* and *E′_s_* are the elasticity moduli of the CFRP and compressive steel bar, respectively; *E_ci_* is the effective elastic modulus of the concrete at the *i*th cycle; $\varepsilon_{si}^{\prime }$ and $\varepsilon_{fi}$ are the longitudinal strains of the compressive steel bar and CFRP at the *i*th cycle; $\varepsilon_{ci}$ is the maximum strain of the compressive concrete at the *i*th cycle; $\varepsilon_{pe}$ is the initial strain of the prestressed CFRP; *A_f_*, *A′_s_*, and *A_s_* are the cross-sectional areas of the CFRP, compressive steel bar, and tensile steel bar, respectively; *f_t_* is the uniaxial tensile strength of the concrete; and $\varepsilon_{t}$ is the uniaxial tensile strain of the concrete.

The relationship between the strains of CFRP and concrete is very important in the above sectional analysis method. However, it is very difficult to calculate precisely the strain of CFRP because of the properties of the bonding interface between the concrete and CFRP. To simplify the analysis, Hui-Huang [12] introduced two extreme CFRP-concrete interfacial states: the fully bonded state and the fully debonded state. In this paper, the fully bonded state was considered because the prestressed CFRP and RC beam were wrapped and bonded together by CFRP strips on the two ends of the prestressed CFRP, which can avoid the debonding failure. For the fully bonded state, the strains along the depth of the strengthened beam are completely compatible, and the plane section assumption can be used. The compatibility equations at a balanced section required that:(8)$$\varepsilon_{fi}=\frac{h_{ci}+a_{i}}{h_{ci}}\varepsilon_{t},$$
(9)$$\varepsilon_{si}=\frac{h_{ci}+a_{i}-c}{h_{ci}}\varepsilon_{t},$$
(10)$$\varepsilon_{ci}=\frac{h-a_{i}-h_{ci}}{h_{ci}}\varepsilon_{t},$$
(11)$$\varepsilon_{si}^{\prime }\;=\frac{h-a_{i}-h_{ci}-c^{\prime }}{h_{ci}}\varepsilon_{t},$$
(12)$$\mathrm{and}\;y_{ci}=h-a_{i}-h_{ci}.$$

By substituting Equations (8)–(12) into Equation (7), all the unknowns at the *i*th cycle can be calculated.

### 2.5. Procedure to Predict the Fatigue Life

The developed fatigue life prediction method was used to predict the fatigue lives of RC beams strengthened with non-prestressed and prestressed CFRP. The step-by-step procedure to implement the model is included in Figure 3. The detailed procedure is as follows:In the procedure, *n_i_* was defined as a unit of circulation and set to 10 cycles in the initial stage. In the steady stage, after 1000 cycles, *n_i_* was set to 1000 cycles. Starting the preliminary cycle, the initial elastic modulus of concrete was *E_c_* and the constitutive relation of the first preliminary cyclic reinforcement was assumed to be linearly elastic.Substituting the geometric dimension, material properties of the specimens, and prestressing level of CFRP into the equilibrium equations (7)–(12), the crack height *a_i_*, the location of the neutral axis *h_ci_*, the strain $\varepsilon_{si}$ of the tensile steel bar, and the compressive strain $\varepsilon_{ci}$ at the top of the concrete at the *i*th cycle under the minimum and maximum loads were calculated.Substituting the value of the stress range $S_{si}$ applied at the tensile steel bar into Equation (3), the fatigue life and accumulated fatigue damage *D* of the specimen under the action of each cycle were obtained.If the accumulated fatigue damage *D* was less than 1, then the effective elastic modulus of concrete *E_ci_* after degradation could be obtained using Equation (4) and (5).According to the strain amplitude of the tensile steel bar in the previous step, the corresponding constitutive model in Equation (6) was selected to calculate the tensile stress $\sigma_{si}$ of the steel bar. Then, $\sigma_{si}$ and *E_ci_* were substituted into Equations (7)–(12) for the next cycle.When the accumulation of fatigue damage *D* = 1, the program was ended, and the fatigue life of the specimen was calculated as $\sum i\times n_{i}$.

## 3. Fatigue Experiments under Cyclic Loading

### 3.1. Specimens and Materials

The typical size and reinforcements (steel bars) of the specimens are shown in Figure 4. The length of all RC beams was 1850 mm, and all had the same cross-sectional dimensions (100 mm width and 200 mm height). The concrete was mixed with cement, water, sand, and gravel in the proportion 1.0:0.5:2.06:3.66. The modulus of elasticity and compressive strength of the concrete were 35.2 GPa and 53.3 MPa, respectively, which were measured using national standard tests (GBJ107-87 and GB/T 50081-2002). The grade HRB 400 steel bars possess an elastic modulus of 206 GPa and a yield strength of 400 MPa. Parameters *k*_1_, *k*_2_, *k*_3_, and *k*_4_ in Figure 1 were chosen as 8.63, 70.5, 1.425, and 72.1, respectively. In this study, the RC beams were strengthened with carbon fiber laminate (CFL) [31] composed of unidirectional carbon fiber silk and epoxide resin for the specimens. Therefore, CFL has the advantages of both carbon fiber plates and sheets. To reduce the influence of multilayer bonding, the size and thickness of the CFL should be designed and knitted according to the project requirements. All CFL used in the test was produced by Toray Advanced Materials Korea Inc., and had a width of 100 mm and normal thickness of 0.23 mm. The elastic modulus and the ultimate tensile strength of the CFL were 230 GPa and 4750 MPa, respectively. A physical diagram of the CFL is shown in Figure 5. The adhesive used between the CFL and concrete was A+B epoxy adhesive, and most of the adhesive penetrated into the concrete. The total thickness of the A+B adhesive layer (with a shear strength of 14 MPa) layer was approximately 0.2 mm.

### 3.2. Prestressing System

In this study, CFL was applied directly via a pretensioned method by jacking up an external reaction frame. The prestressed CFL was pasted onto an RC beam with A+B epoxy resin in this method. When the A+B epoxy resin was completely cured, the prestressing system was released. Extra anchorage needed to be provided if the prestressing level was high. The advantage of the pretensioned method was that the RC beam could be kept intact, rather than drilling holes in the specimen. The CFL was stretched by the pretensioned method described above, and the CFL was wrapped at both ends of the beam to avoid peeling failure. The tensile prestress and bonding procedures of the CFL can be subdivided into four processes: tensioning, bonding, curing, and releasing. We developed the prestressing system for stretching CFL and reinforcing RC beams, the details and processes of which are shown in Figure 6 and are specifically described in the reference [31].

### 3.3. Experimental Method

A total of 14 specimens were produced in this study. Prestressing levels of 0% and 10% of were applied to the CFL strengthening the RC beams in the experimental study. A level of 0% indicats no prestressing. The RC beams strengthened with a 10% prestress level (475 MPa) were wrapped with CFL on both ends. Table 1 shows all details of the specimens.

#### 3.3.1. Testing Procedure

We employed a loading method of three-point bending. The experiments were carried on Material Testing System (MTS) with a total capacity of 100 kN in the forced mode. Static testing was carried out before the fatigue experiment, and the ultimate loads of specimens strengthened with non-prestressed and 10% prestressed CFL were 51.5 kN and 62.5 kN, respectively. The stress ratio *R* was 0.2, and the frequency of constant amplitude load was 10 Hz. The experimental conditions are summarized in Table 1.

#### 3.3.2. Measurements

During the tests, a dynamic strain indicator was used to measure the strains, and the data acquisition frequency was set to 100 Hz. The evolution of the crack shape was surveyed by employing a 3-dimensional digital image correlation (DIC) method [31], which is an optical measurement technique with does not need surface contact to monitor the cracks. In the test settings, the side surface of the specimen was monitored using the DIC system (Figure 7). Marking a pattern of black spots on a white background (Figure 8), the displacements and strain figures surrounding the main crack were measured at succedent loading steps. Moreover, some crack characteristics such as the crack height, width and spacing were monitored.

### 3.4. Experimental Results

Table 1 shows the fatigue lives of all specimens. Because the CFRP wrapped around the ends of the RC beam can avoid debonding failure, so the failure mode of all the specimens was tensile steel bar fracture, as shown in Figure 9. The tensile steel bar fractured at the main cracked section after substantial fatigue damage accumulation. Then, the tensile force carried by the steel bar was transferred to the CFL, which resulted in the fracture of the CFL.

During the fatigue tests, the crack shape evolution was surveyed by the DIC system. Figure 10 shows the crack propagation under different loading cycles. Figure 10 also shows the relationships between the numbers of loading cycles and the main crack heights of specimens of different prestressing levels. The observed crack growth on the specimen can be divided into three stages, namely, fast, stable, and unstable propagation stages. During the first stage, cracks occurred, and one of them developed rapidly into the main crack. After the first stage, the observed changes in fatigue damage had become extremely small over a long period of time. In the stable propagation stage, approximately 95% of the total fatigue life of the specimen was the period of main crack propagation life. Therefore, in engineering practice, the full fatigue lives of the RC beams could be approximately predicted by the fatigue lives of the main crack propagation stage. The tensile steel bars would fracture at the main crack section after a large amount of fatigue damage accumulation. The final stage lasted a relatively short time.

### 3.5. Strain Response

P0-30-1, P10-30-1, and P10-35-1 were CFL-strengthened specimens with different prestressing levels and were subjected to different peak loads. Figure 11 shows the strains of the tensile steel bars with respect to the loading cycles at the peak loads. As shown in Figure 12, the strains in the main steel bars of each specimen experienced a significant increase and then increased more slowly during the remaining load cycles. Compared with RC beams strengthened with non-prestressed CFL, prestressed CFL could reduce the strains of main steel bars in an obvious way. From the beginning to the final fracture, the stress increments in the main steel bars were basically the same.

## 4. Model Verification

### 4.1. Crack Propagation

In this paper, the experimental results (*a*-*N* curves) mentioned in the Section 3.4 were used for verification of the model. All the geometric dimensions, material properties of specimens, and prestressing levels of the CFL mentioned in the Section 3.4 were substituted into fatigue prediction model to obtain the main crack height, *a*, after each cycle until the cumulative damage reached *D* = 1. The theoretical values of crack height were then compared with the experimental data, as shown in Figure 12.

From the comparison results, we can see that the theoretical values were in good agreement with the experimental data, with an average error of 1.3%. As shown in Figure 13, prestressing could greatly improve the resistance to fracture. The theoretical model also described a high expansion rate of the main crack in the first several loading cycles, and the crack height then increased only slightly over most of the fatigue process.

### 4.2. Fatigue Life Prediction

To further verify the fatigue life prediction model proposed in this study, more test results were needed. Fortunately, we has previously carried out fatigue tests on CFL-reinforced RC beams with various prestressing levels (15% and 22%) [14]. The geometric dimensions and material properties of the specimens were the same as in this study.

If the gradual degradation of the performance of the concrete was not considered, the fatigue lives of the specimens could be predicted using Equation (3). The stress ranges of the tensile steel bars in Equation (3) were calculated using the static Equations (7)–(12) directly. To compare the two models, one considering the degradation of concrete and the other not, four different prestressing levels (0%, 10%, 15%, and 22%) of CFL-reinforced RC beams (summing to 30 specimens) were involved.

Figure 14a shows the comparison of prediction lives calculated by the accumulative damage model without considering the degradation of concrete performance against the experimental data. The predicted lives calculated by the accumulative damage model considering the degradation of concrete performance were compared with the experimental data, as shown in Figure 14b. It was found that the fatigue lives calculated without considering the degradation of concrete performance were larger than those in the experimental data, and the average relative error was 33%; However, the fatigue lives calculated with the degradation of concrete performance were closer to the experimental data, the average relative error was 10%. This shows that taking into account the gradual degradation of the concrete performance, which more truly describes the influence of the stress increase of the main steel bar on the fatigue life of the strengthened concrete, could reduce the prediction error effectively.

## 5. Conclusions

Considering the gradual degradation of concrete performance, a fatigue life prediction model for RC beams strengthened with prestressed or non-prestressed CFRP was proposed. Moreover, an experimental study was also carried out to probe into the fatigue behavior of non-prestressed and prestressed CFL-reinforced RC beams. During the test, the fatigue crack patterns were captured using a 3D-DIC system, and the fatigue lives of a total of 30 strengthened beams were obtained. By a comparison between the predicted lives and experimental data, the effectiveness of the proposed model was verified. The following conclusions could be drawn from the experimental and theoretical analysis results presented in this paper:The proposed accumulative damage model can well describe the crack height and stress range of tensile steel bars at different loading cycles. The fatigue lives of non-prestressed and prestressed CFRP-reinforced RC beams can also be predicted by the proposed model.The predicted results indicated that the predicted fatigue lives considering the gradual degradation of the concrete performance were close to the experimental data with an average relative error of 10%. However, the average relative error of predicted fatigue lives without considering the gradual degradation of concrete performance was 33%. The proposed model could accurately and reliably predict the flexural fatigue life of non-prestressed and prestressed CFRP-reinforced RC beams.The analysis results indicated that prestressed CFRP could decrease the stress of main steel bars in RC beams and improved the fatigue performance of the beams effectively. Compared with non-prestressed CFL-reinforced RC beams, the fatigue lives of prestressed CFL-reinforced RC beams (to a 10% prestressing level) were greatly increased.

This research provided a theoretical model to describe fatigue behavior of reinforced concrete structure. The availability and applicability of this model should be further verified in similar reinforce concrete structure.

## Figures and Tables

**Figure 1 polymers-11-00130-f001:**
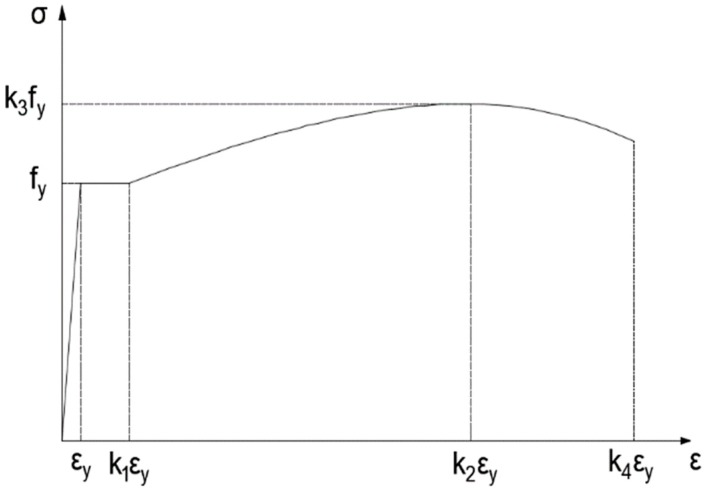
Stress-strain relationship for tensile steel bars.

**Figure 2 polymers-11-00130-f002:**
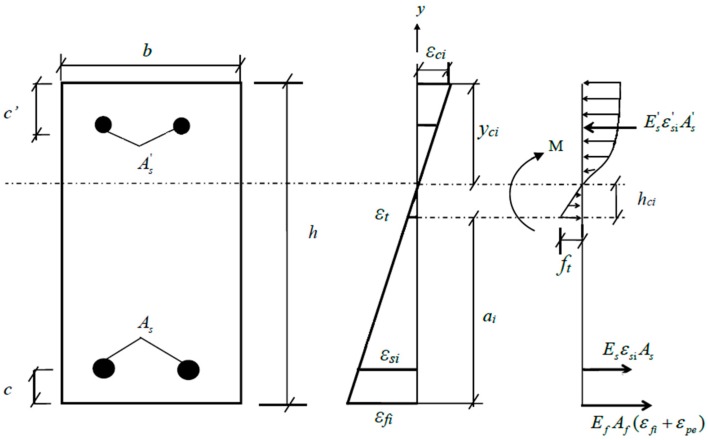
Stress-strain distribution at the main cracked section.

**Figure 3 polymers-11-00130-f003:**
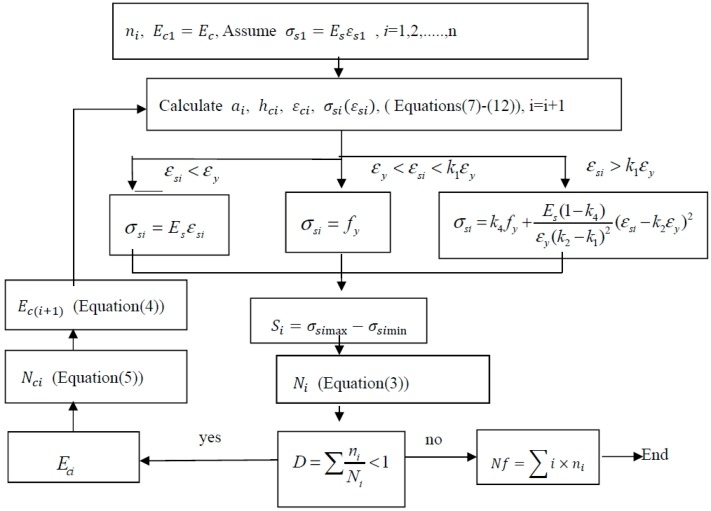
Procedure for calculating the fatigue life *N_f_*.

**Figure 4 polymers-11-00130-f004:**
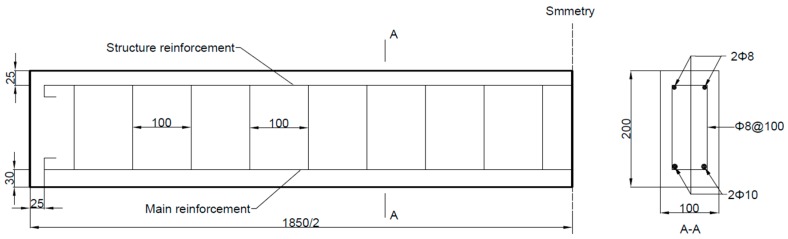
Size and reinforcements (steel bars) in the reinforce concrete (RC) beam (unit: mm).

**Figure 5 polymers-11-00130-f005:**
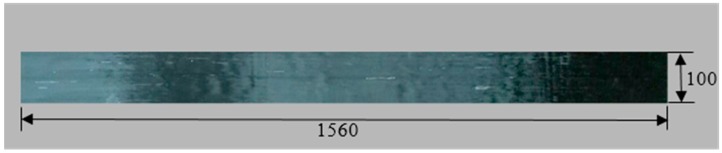
Picture of real products (carbon fiber laminate (CFL)).

**Figure 6 polymers-11-00130-f006:**
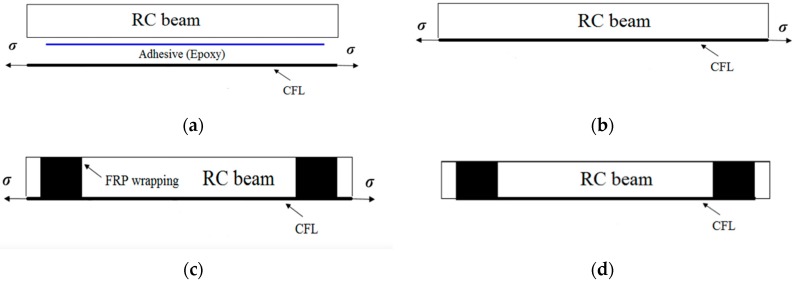
Strengthening method with prestressed CFL for RC beams. (**a**) Tensioning; (**b**) Bonding; (**c**) Curing; (**d**) Releasing.

**Figure 7 polymers-11-00130-f007:**
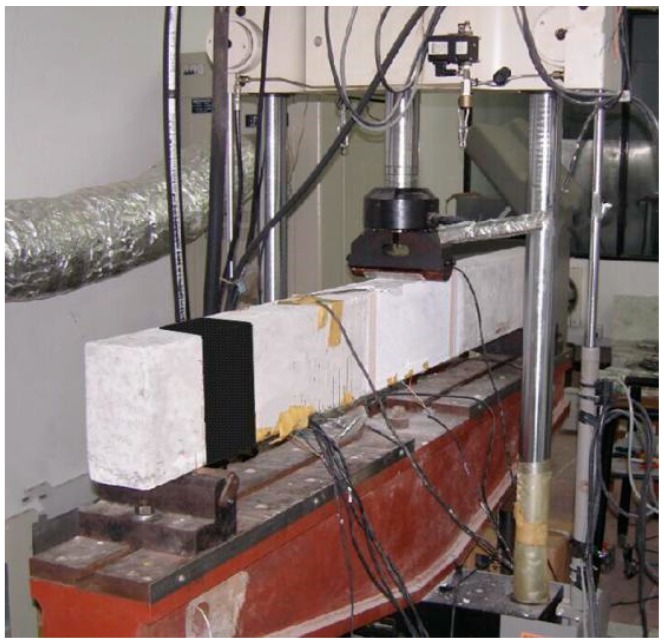
Material Testing System (MTS).

**Figure 8 polymers-11-00130-f008:**
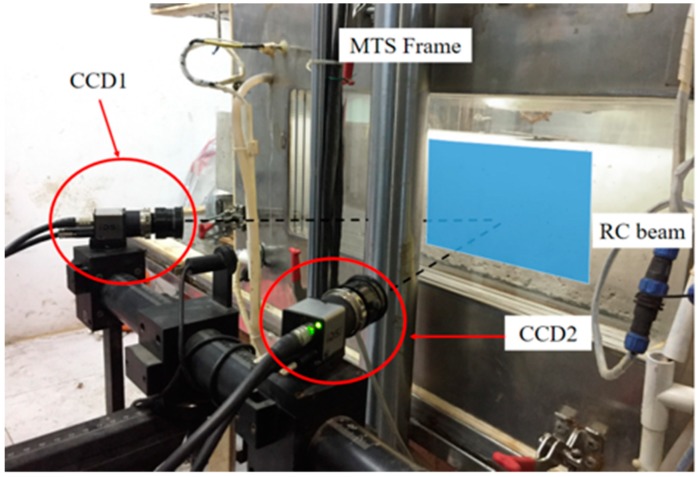
3D-DIC system.

**Figure 9 polymers-11-00130-f009:**
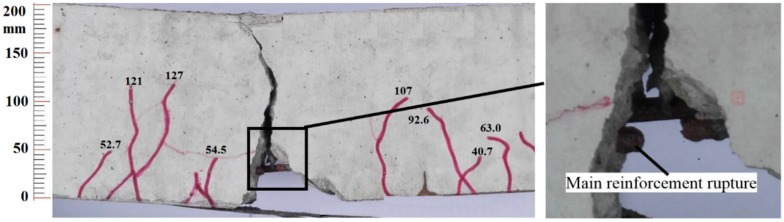
Fatigue failure mode (main steel bar fracture) of the strengthened beam (unit: mm).

**Figure 10 polymers-11-00130-f010:**
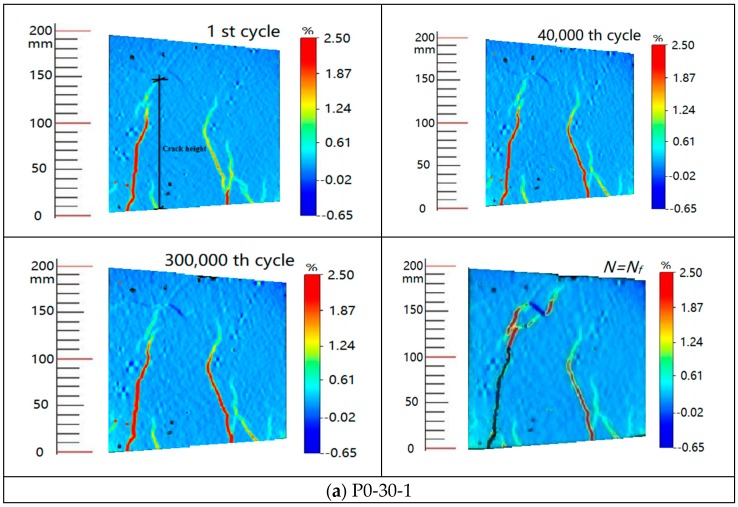
Crack pattern evolution at different loading cycles. (**a**) P0-30-1; (**b**) P10-40-2.

**Figure 11 polymers-11-00130-f011:**
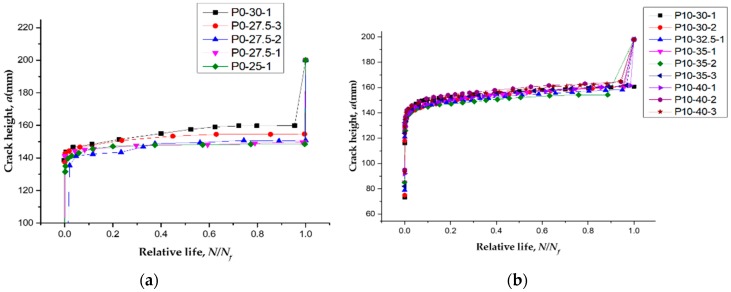
Experimental *a*-*N* curves for specimens with different prestressing levels. (**a**) RC beams strengthened with non-prestressed CFL; (**b**) RC beams strengthened by CFL with a prestressing level of 10%.

**Figure 12 polymers-11-00130-f012:**
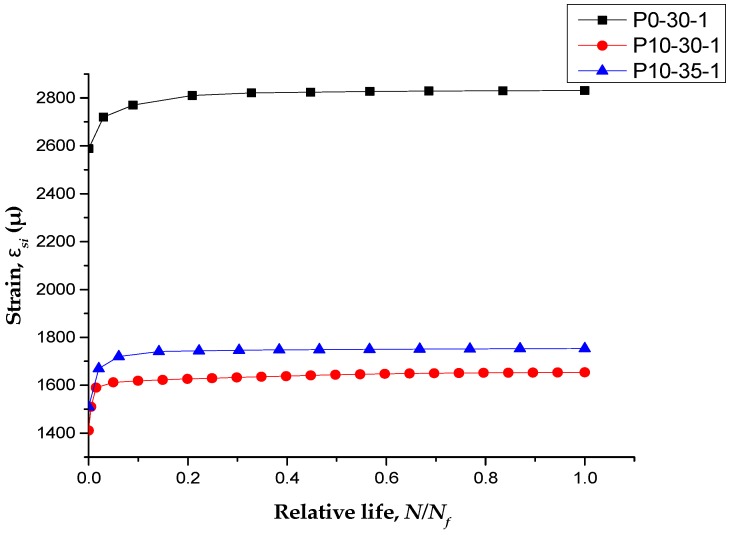
Strains of main steel bars at the peak loads.

**Figure 13 polymers-11-00130-f013:**
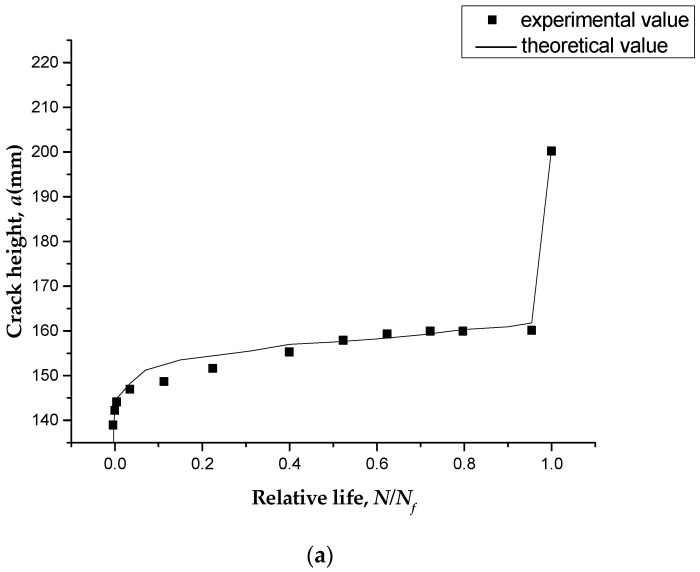
Comparison between *a*-*N* curves of experimental data and predicted values. (**a**) P0-30-1; (**b**) P10-30-1; (**c**) P10-40-1.

**Figure 14 polymers-11-00130-f014:**
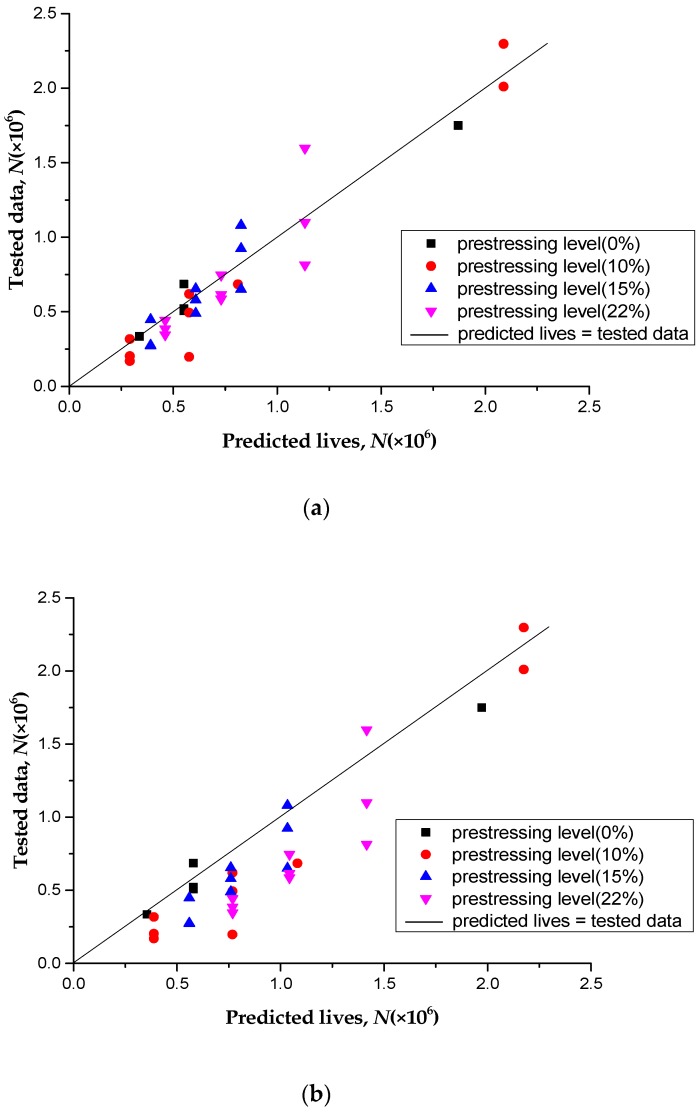
Tested data versus predicted lives for two models: (**a**) Considering the degradation of concrete performance; (**b**) Without considering the degradation of concrete performance.

**Table 1 polymers-11-00130-t001:** Experimental conditions and testing results.

Specimen No.	Prestressing Levels	Peak Loads, *P*_max_ (kN)	Minimum Loads, *P*_min_ (kN)	Initial Crack Height (mm)	Final Crack Height (mm)	Initia Steel Strain (μ)	Fatigue Lives, *N_f_* (Cycles)
P0-25-1	0%	25.0	5.0	139.9	148.6	1140.8	1750067
P0-27.5-1		27.5	5.5	142.9	154.7	1467.6	507341
P0-27.5-2				141.1	150.9	1468.4	686534
P0-27.5-3				141.2	149.7	1466.2	521580
P0-30-1		30.0	6.0	144.0	159.9	1642.2	335327
P10-30-1	10%	30.0	6.0	138.5	160.8	1128.6	>2000000
P10-30-2				136.6	161.0	1128.0	2296825
P10-32.5-1		32.5	6.5	139.9	158.2	1249.0	684595
P10-35-1		35.0	7.0	140.3	161.0	1368.7	494496
P10-35-2				137.4	154.2	1368.1	197776
P10-35-3				137.6	161.7	1369.4	620020
P10-40-1		40.0	8.0	136.1	161.5	1609.2	203074
P10-40-2				141.8	164.6	1607.6	317900
P10-40-3				141.1	165.0	1609.9	169990

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
