# Peer review of "Fatigue Life Prediction of Reinforced Concrete Beams Strengthened with CFRP: Study Based on an Accumulative Damage Model"

_polymers, 2019, doi:10.3390/polym11010130_

Round 1

Reviewer 1 Report

The Reviewer acknowledges the Authors’ efforts in revising the manuscript. I have carefully gone through the review comments and corresponding comments. In my technical opinion, paper needs minor modifications before publication.

1.     In my opinion, related papers from MDPI journals should be cited. A few are appended below for adequately referencing

   i.         Hanif, A.; Kim, Y.; Park, C. Numerical Validation of Two-Parameter Weibull Model for Assessing Failure Fatigue Lives of Laminated Cementitious Composites—Comparative Assessment of Modeling Approaches. Materials 2019, 12, 110.

                       ii.         Park, H.B.; Park, J.-S.; Kang, J.-Y.; Jung, W.-T. Fatigue Behavior of Concrete Beam with Prestressed Near-Surface Mounted CFRP Reinforcement According to the Strength and Developed Length. Materials 2019, 12, 51.

2.     Regarding the first reference in the first comment, the Authors should see that a modelling software is used for fatigue life prediction. How do the Authors see its applicability in the present research which is currently submitted by the Author? It is worth mentioning that the opensource program ProFatigue® has been developed several years ago which employed three Weibull and Gumbell models for fatigue life prediction. The conclusions section should better be enhanced accordingly, as well

3.     Regarding the second reference in the first comment, the Authors have not included it in the literature survey. What’s the difference between the two?

4.     Statistical information on data processing is still not provided yet. Kindly provide.

5.     Figure 10; the images are still of low quality due to reduced pixels. Please rearrange the sub-figures appropriately and provide high quality images. Upon publication, the images shall lose the desired quality leading to difficulties in data interpreting by a potential reader.

6.     The numbers mentioned on the specimen in Figure 9 are illegible. Please enlarge the font.

7.     Nomenclature should be placed at the start of the manuscript, rather at the end.

Author Response

Response to Reviewer 1 Comments

The authors would like to thank the reviewers and the editors for their constructive comments. All comments have been carefully considered and, wherever appropriate, revisions have been made to the manuscript. Responses to the comments and revisions implemented in the paper are detailed below.

Point 1: In my opinion, related papers from MDPI journals should be cited. A few are appended below for adequately referencing

 i.  Hanif, A.; Kim, Y.; Park, C. Numerical Validation of Two-Parameter Weibull Model for Assessing Failure Fatigue Lives of Laminated Cementitious Composites—Comparative Assessment of Modeling Approaches. Materials 2019, 12(1), 110; https://doi.org/10.3390/ma12010110

ii.  Park, H.B.; Park, J.-S.; Kang, J.-Y.; Jung, W.-T. Fatigue Behavior of Concrete Beam with Prestressed Near-Surface Mounted CFRP Reinforcement According to the Strength and Developed Length. Materials 2019, 12(1), 51; https://doi.org/10.3390/ma12010051

Response 1: I read the related papers(i,ii) carefully. The two papers are cited in references, and the introduction corresponding to the two papers are added in line 65-70.

Point 2: Regarding the first reference in the first comment, the Authors should see that a modelling software is used for fatigue life prediction. How do the Authors see its applicability in the present research which is currently submitted by the Author? It is worth mentioning that the opensource program ProFatigue® has been developed several years ago which employed three Weibull and Gumbell models for fatigue life prediction. The conclusions section should better be enhanced accordingly, as well.

Response 2: I read the first reference carefully. Although the program ProFatigue® is very useful to predict the fatigue lives of materials and can significantly reduce the computational effort and time, the limitation of the program is that the software is designed for isotropic materials. For complex composite structure, my research provided a theoretical model to predict fatigue life based on the fatigue behavior of steel bar. My research described the fatigue behavior of complex structure from mechanics, and its applicability should be wide. The research need to be further verified in similar reinforced concrete structure. The additional explanation of the model is added in the end.

Point 3: Regarding the second reference in the first comment, the Authors have not included it in the literature survey. What’s the difference between the two?

Response 3: The literature survey of second reference was added in introduction. In the second reference, a fatigue test was carried out to study the fatigue behavior of RC beam strengthened with prestressed NSMR. In my study, I proposed a theoretical analysis model to predict the fatigue behavior of RC beam strengthened with non-prestressed and prestressed CFRP. The fatigue test in my paper is used to verify this model. The method in my model should be applied to similar reinforced concrete structure.

Point 4: Statistical information on data processing is still not provided yet. Kindly provide.

Response 4: Statistical information (such as crack height and steel strain) is provided in table 1.

Point 5: Figure 10; the images are still of low quality due to reduced pixels. Please rearrange the sub-figures appropriately and provide high quality images. Upon publication, the images shall lose the desired quality leading to difficulties in data

Response 5: To improve the quality of images, the sub-figures were rearranged and handled.

Point 6: The numbers mentioned on the specimen in Figure 9 are illegible. Please enlarge the font.

Response 6:The numbers mentioned on the specimen in Figure 9 were enlarged. 

Point 7: Nomenclature should be placed at the start of the manuscript, rather at the end.

Response 7: I read some papers from ELSEVIER journals and find nomenclatures are placed either on the start or on the second page of the manuscript. I didn’t find the required format of the nomenclature in MDPI journals, so nomenclature was placed at the start of second page for the time being.

Reviewer 2 Report

the paper is significantly improved.

Author Response

Thank the reviewer for your reading and comments. Some minor spelling errors in the paper have been checked and corrected.

Round 2

Reviewer 1 Report

The paper is well revised and can be accepted in current form. I have no further comment.

This manuscript is a resubmission of an earlier submission. The following is a list of the peer review reports and author responses from that submission.

Round 1

Reviewer 1 Report

Paper review: Fatigue life prediction of RC beams strengthened with non-prestressed and prestressed CFRP based on accumulative damage model

In this study, a fatigue life prediction model of RC beams strengthened with prestressed CFRP was developed based on accumulative damage model. The gradual degradation of the performance of the concrete was considered in the fatigue life prediction model. In addition, experimental investigations were carried out to verify the proposed model. Based on the laboratory test results, the authors concluded that the predicted main crack propagation curves and the fatigue lives were close to the experimental data. This research also showed that the prestressed CFRP could reduce the stress level in the steel bars in RC beams and improve the fatigue performance of the beams.

I have looked over this manuscript carefully. This paper deals with a very important topic in our field. The prestressed CFRP has been used increasingly in recent decades. The research results will bring great impacts to the design, maintenance, and evaluation of civil infrastructures. Therefore, I think this paper can be accepted after a minor revision. 

1. I can understand the authors’ writing; however, I still feel the English of this paper is quite low. The authors should substantially improve the paper’s English quality.

2. Since this paper contains many equations, a list of symbols/notations/nomenclature should be added.

3. The figure quality needs to be improved, e.x., figure 2. Figure 3. The broken line needs be fixed.

4. The authors should show the R^2 ratios, so that we can know how good the prediction is.

Author Response

Response to Reviewer 1 Comments

The authors would like to thank the reviewers and the editors for their constructive comments. All comments have been carefully considered and, wherever appropriate, revisions have been made to the manuscript. Responses to the comments and revisions implemented in the paper are detailed below.

 Point 1:  I can understand the authors’ writing; however, I still feel the English of this paper is quite low. The authors should substantially improve the paper’s English quality

Response 1:The manuscript was polished at https://www.mdpi.com/authors/english..

 Point 2: Since this paper contains many equations, a list of symbols / notations / nomenclature should be added.

Response 2:Thanks. As the reviewer suggested, a list of nomenclature was added in page 2.

 Point 3: The figure quality needs to be improved, e.x., figure 2. Figure 3. The broken line needs be fixed.

Response 3: Thanks. As the reviewer suggested, the broken line in Figure 2 and Figure 3 were fixed. Furthermore, the quality Figure 4 was improved.

Point 4: The authors should show the R^2,so that we can know how good the prediction is.

Response 4: The line in Figure 14 was not a fitting line. If the point was on the line, which meant the predicted lives equal to tested data of specimen. The average relative error could show accuracy of prediction.

Reviewer 2 Report

This paper investigates the fatigue behavior of RC beams strengthened with CFRP laminates. The paper exhibits major issues concerning the methodology, manuscript organization, and results analysis. There are major issues in the paper which must be addressed prior its acceptance. References are inadequate and the results discussion is insufficient to convey the meaning. A major concern Is the very high similarity index (>31%). Major text is directly copied from [1] which is around 10%.

 I am appending below the major and minor comments for Authors’ consideration.

1.     Please rephrase the text to reduce the similarity, throughout.

2.     Title should be revised. Use of abbreviations is not encouraged in title. Also, it’s not appropriate to use the terms “non-prestressed and prestressed” in the title.

3.     Line 29-31; “Bonding the carbon fiber …… past decades”, please provide references.

4.     Please consider citing the following references relating to different fatigue models [2–6].

5.     Line 224-225;” The modulus of elasticity and compressive strength of the concrete cube was 35.2 GPa and 53.3 MPa, respectively.” How it was determined? Is it published earlier by the same Authors?

6.     Pictures / photos of CFL must be provided. Also, mention the suppliers. The properties of CFL have been determined by the Authors or provided by the manufacturer?

7.     Figure 4; please indicate the measurement units.

8.     A critical concern is that only one stress ratio was evaluated (0.2) which makes the modelling less accurate. Kindly justify. Further, the stress ratio is based on the static test result which is not explained here. It should also be covered in this paper.

9.     What’s the reason for using 10 Hz loading frequency? Also, I would advise to graphically show the frequency and corresponding deformation.

10.  Figure 8; please provide the scale bar.

11.  Only one beam per specimen is tested, which in my opinion is not adequate. What about the data reliability? Evaluating just one sample is scientifically sound? Provide justification with reference to the published findings in similar area.

12.  Figure 9; scale bar?

13.  Figure 11, the units of strain are incorrect. Strain is expressed in mm/mm or %. Please correct.

14.  For all the graphs, please express the unit in parenthesis. Currently, the Authors have used slash which is not appropriate.

15.  Figure 13; parameters and coefficient of correlation for the straight line must be shown.

REFERENCES

1.        Guo, X.; Yu, B.; Huang, P.; Zheng, X.; Zhao, C. J-integral approach for main crack propagation of RC beams strengthened with prestressed CFRP under cyclic bending load. Engineering Fracture Mechanics 2018, 200, 465–478, doi:10.1016/j.engfracmech.2018.08.003.

2.        Hanif, A.; Usman, M.; Lu, Z.; Cheng, Y.; Li, Z. Flexural Fatigue Behaviour of Thin Laminated Cementitious Composites Incorporating Cenosphere Fillers. Materials & Design 2018, 140, 267–277, doi:10.1016/j.matdes.2017.12.003.

3.        Huang, B.-T.; Li, Q.-H.; Xu, S.-L.; Liu, W.; Wang, H.-T. Fatigue deformation behavior and fiber failure mechanism of ultra-high toughness cementitious composites in compression. Materials & Design 2018, 157, 457–468, doi:10.1016/j.matdes.2018.08.002.

4.        Huang, B.-T.; Li, Q.-H.; Xu, S.-L.; Zhou, B.-M. Frequency Effect on the Compressive Fatigue Behavior of Ultrahigh Toughness Cementitious Composites: Experimental Study and Probabilistic Analysis. Journal of Structural Engineering 2017, 143, 04017073, doi:10.1061/(ASCE)ST.1943-541X.0001799.

5.        Huang, B.-T.; Li, Q.-H.; Xu, S.-L.; Zhou, B.-M. Tensile fatigue behavior of fiber-reinforced cementitious material with high ductility: Experimental study and novel P - S - N model. Construction and Building Materials 2018, 178, 349–359, doi:10.1016/j.conbuildmat.2018.05.166.

6.        Lu, C.; Dong, B.; Pan, J.; Shan, Q.; Hanif, A.; Yin, W. An investigation on the behavior of a new connection for precast structures under reverse cyclic loading. Engineering Structures 2018, 169, 131–140, doi:10.1016/j.engstruct.2018.05.041.

Author Response

Response to Reviewer 2 Comments

The authors would like to thank the reviewers and the editors for their constructive comments. All comments have been carefully considered and, wherever appropriate, revisions have been made to the manuscript. Responses to the comments and revisions implemented in the paper are detailed below.

Point 1: Please rephrase the text to reduce the similarity, throughout.

Response 1: The similarities throughout the paper were checked and replaced.

Point 2: Title should be revised. Use of abbreviations is not encouraged in title. Also, it’s not appropriate to use the terms “non-prestressed and prestressed” in the title.

Response 2: Thanks. I think the title will appear too long if abbreviations are not used. Furthermore, CFRP is common and well known in many fields.

Point 3: Line 29-31; “Bonding the carbon fiber …… past decades”, please provide references.

Response 3: References [1-2] were provided for “Bonding the carbon fiber …… past decades”.

Point 4: Please consider citing the following references relating to different fatigue models [2–6].

Response 4: Following references [2, 4] were cited in references [15, 16]. 

Point 5: Line 224-225;” The modulus of elasticity and compressive strength of the concrete cube was 35.2 GPa and 53.3 MPa, respectively.” How it was determined? Is it published earlier by the same Authors?

Response 5: The modulus of elasticity and compressive strength of the concrete were measured by national standard tests (GBJ107-87 and GB/T 50081-2002). The explanation was added in line 229-230.

Point 6: Pictures / photos of CFL must be provided. Also, mention the suppliers. The properties of CFL have been determined by the Authors or provided by the manufacturer?

Response 6: The photo of CFL was added in Figure 5. Also, the suppliers was mentioned in line 236. The properties of CFL was provided by the manufacturer.

Point 7: Figure 4; please indicate the measurement units.

Response 7: The measurement units of Figure 4 was indicated at the end of title.

Point 8: A critical concern is that only one stress ratio was evaluated (0.2) which makes the modelling less accurate. Kindly justify. Further, the stress ratio is based on the static test result which is not explained here. It should also be covered in this paper.

Response 8: The modelling does not depend on the fatigue experimental datathis is the significance and innovation of this paper, so it was suitable for any stress ratio instead of only 0.2.  As a check, stress ratio 0.2 (most frequently used in fatigue test) was used to prove the modelling. The static test result was a guide to choose the maximum load, so the static test result was added in line 269.  

Point 9: What’s the reason for using 10 Hz loading frequency? Also, I would advise to graphically show the frequency and corresponding deformation.

Response 9: The frequency range of Material Testing System (MTS) 810 is in 0Hz-20 Hz. Past experience showed that the frequency was set 10Hz can avoid resonance during test.

Point 10: Figure 8; please provide the scale bar.

Response 10: Thanks. As the reviewer suggested, the scale bar was added in Figure 9. (Because a figure was added, then the Figure 8 was turned into Figure 9.)

Point 11: Only one beam per specimen is tested, which in my opinion is not adequate. What about the data reliability? Evaluating just one sample is scientifically sound? Provide justification with reference to the published findings in similar area.

Response 11: There are about three specimens at each condition. The specimens were used to verify the model instead of fitting, so the number of specimens was enough.

Point 12: Figure 9; scale bar?

Response 12: The scale bar was added in Figure 10. 

Point 13: Figure 11, the units of strain are incorrect. Strain is expressed in mm/mm or %. Please correct.

Response 13: Thanks. As the reviewer suggested, the units of strain were corrected in Figure 12.

Point 14: For all the graphs, please express the unit in parenthesis. Currently, the Authors have used slash which is not appropriate.

Response 14: The units of Figure 11, 12, 13 were expressed in parenthesis.

Point 15: Figure 13; parameters and coefficient of correlation for the straight line must be shown.

Response 15: The line in Figure 14 was not a fitting line. If the point was on the line, which meant the predicted lives equal to tested data of specimen. The meaning of the line was added and shown in Figure14.

Round 2

Reviewer 2 Report

I have found that Authors have taken the review comments very lightly. Please re-read the comments from first review round. I am appending a few more comments to improve the paper.

1. In the first Review round, I suggested to cite six references which are related to the subject area, but the Authors cited two of them while mentioned in the response that all papers have been cited. Please check the details of remaining four from comments of first review and revise accordingly.

2. Title revision suggested: Fatigue life prediction of reinforced concrete beams strengthened with CFRP: Study based on accumulative damage model.

3. Figure 2; plz cite reference in caption for the model (Whitney stress block?)

4. Figure 5 is incomprehensible without scale bar. Please provide within the figure.

5. Figure 9 (previously Figure 8 in original manuscript) scale bar missing while the Authors emphasize in respires that the scale is provided.

6. Figure 10; low quality images. Must be revised.

7. As the Authors mentioned in the response report that three samples were tested for each category, please provide the statistical analysis of data.